# Functional Dissection of P1 Bacteriophage Holin-like Proteins Reveals the Biological Sense of P1 Lytic System Complexity

**DOI:** 10.3390/ijms23084231

**Published:** 2022-04-11

**Authors:** Agnieszka Bednarek, Agata Cena, Wioleta Izak, Joanna Bigos, Małgorzata Łobocka

**Affiliations:** 1Institute of Biochemistry and Biophysics of the Polish Academy of Sciences, 02-106 Warsaw, Poland; a.kielan@ibb.waw.pl (A.B.); agatacena@yahoo.com (A.C.); wioleta.woznica@gmail.com (W.I.); 2Autonomous Department of Microbiology, Faculty of Agriculture and Biology, Warsaw University of Life Sciences, SGGW, 02-776 Warsaw, Poland; nasierowskajoanna@gmail.com

**Keywords:** bacteriophage P1, bacterial cell lysis, holin, pinholin, endolysin, SAR-endolysin, *Escherichia coli*, *Enterobacter cloacae*

## Abstract

P1 is a model temperate myovirus. It infects different *Enterobacteriaceae* and can develop lytically or form lysogens. Only some P1 adaptation strategies to propagate in different hosts are known. An atypical feature of P1 is the number and organization of cell lysis-associated genes. In addition to SAR-endolysin Lyz, holin LydA, and antiholin LydB, P1 encodes other predicted holins, LydC and LydD. LydD is encoded by the same operon as Lyz, LydA and LydB are encoded by an unlinked operon, and LydC is encoded by an operon preceding the *lydA* gene. By analyzing the phenotypes of P1 mutants in known or predicted holin genes, we show that all the products of these genes cooperate with the P1 SAR-endolysin in cell lysis and that LydD is a pinholin. The contributions of holins/pinholins to cell lysis by P1 appear to vary depending on the host of P1 and the bacterial growth conditions. The pattern of morphological transitions characteristic of SAR-endolysin–pinholin action dominates during lysis by wild-type P1, but in the case of *lydC lydD* mutant it changes to that characteristic of classical endolysin-pinholin action. We postulate that the complex lytic system facilitates P1 adaptation to various hosts and their growth conditions.

## 1. Introduction

The lytic development of bacteriophages in bacterial cells ends with cell lysis to release the progeny phages. The effectiveness of lysis and its precise control in time is crucial for the evolutionary fate of a given phage [1]. Premature lysis leads to the release of immature, noninfective phages. Delayed lysis leads to failure in evolution with competing phages.

Typically, double-stranded DNA phages encode two kinds of essential proteins to lyse cells at the release of phage progeny (Figure 1). One of them, endolysin, destroys the cell wall by digesting chemical bonds essential for the integrity of cell wall peptidoglycan (PG). Most phages produce endolysins that are unable to cross the cytoplasmic membrane (CM) and require holins, proteins that form large, nonspecific membrane pores (holes) [2,3,4]. In the classical paradigm, endolysin is free in the cytoplasm and the holin, dispersed in the cellular membrane, is triggered to assemble into large rafts, which create very large sites of membrane disruption (Figure 1A). This provides fully active endolysins with access to the cell wall peptidoglycan, which results in the local disruption of the cell wall envelope and escape of the cell contents. As an exception to this classical paradigm, some phages were found to produce endolysins that can cross the cytoplasmic membrane without the help of canonical holins [5,6,7,8,9,10]. In the phages of Gram-negative bacteria, such endolysins contain at their N-terminus a so-called signal arrest release (SAR) domain, a non-cleavable type II signal anchor [3,9,10,11,12,13,14,15,16]. Thus, in a recently described second paradigm, the inactive SAR-endolysin anchored to the cytoplasmic membrane is released and folded to its active form when triggered by the depolarization of the membrane by so-called pinholin, a small holin-like protein that can form a multitude of nm scale holes (pinholes) in the CM (Figure 1B) [6,17,18,19,20,21].

Despite the different mechanisms of action, the triggering of lysis by either holins or pinholins is associated with the collapse of proton motive force (PMF) and cessation of macromolecular synthesis, leading to cell death [9,23,24]). Thus, in addition to their role in the liberation of endolysins or SAR-endolysins to the cell wall, holins/pinholins are postulated to be essential for cell lysis by causing cell death. This, in turn, would allow lysis to occur due to the accumulation of endolysin-mediated cell wall damage, which otherwise would be readily repaired in rapidly growing cells [24].

The triggering of cell lysis occurs precisely when mature phages are ready to be released. It is associated with the achievement of a threshold concentration of holin/pinholin in the cytoplasmic membrane, leading to the formation of holes/pinholes and the release or activation of endolysin/SAR-endolysin [20,25] (reviewed in [26]). In certain phages, the timing of lysis triggering is influenced by so-called antiholins, proteins that act by forming membrane complexes with holins or regulating holin function in another, unknown way (reviewed in [9]). Additionally, the phages of Gram-negative bacteria encode so-called spanins, proteins that destroy the outer membrane (OM) of lysing cells by spanning it with the cytoplasmic membrane through the endolysin/SAR-endolysin-mediated damage of the cell wall [27,28,29,30]. Most often, spanins are either two-component complexes of OM lipoprotein (o-spanin) and a type II inner membrane (IM) protein (i-spanin), although some are produced in the form of a single polypeptide of each of these functions (reviewed in [9]). In laboratory cultures grown with shaking, spannins are not essential for lysis.

Typically, phages encode one set of lytic proteins, and typically endolysin, holin, and antiholin are encoded by the same operon. The antiholin gene is often embedded within the holin gene, and spanin genes are immediately downstream, as in the case of λ phage [31,32,33,34,35,36,37,38]. P1, a model temperate bacteriophage of known genomic sequence (93.6 kb) and wide host range among enterobacteria, is atypical in that respect [39,40,41]. Three P1 proteins, a prototypical SAR–endolysin Lyz of lysozyme activity, holin LydA, and antiholin LydB, have been shown to cooperate in the lysis of *E. coli* when produced from their genes cloned in a plasmid under the control of the inducible promoter [5,41]. Thus, LydA has been considered a primary holin of P1. However, Lyz is encoded by a late operon, unlinked to the late *dar* operon, which encodes LydA and LydB and also antirestriction and capsid morphogenesis-associated functions (Figure 2) [40,42,43,44,45,46,47,48]. Moreover, in addition to LydA, the P1 genome encodes two other proteins with predicted holin features, LydC and LydD [40]. LydC (93 aa) is encoded by a monocistronic operon preceded by the putative early promoter and immediately preceding the *dar* operon (Figure 2). LydD (84 aa) is encoded by the second gene of the *lyz* operon, as would be expected for a pinholin. Its amino acid sequence differs from those of LydA and LydC, which are 29% identical to each other. However, like LydA and LydC, LydD contains two predicted transmembrane helices, and its C-terminus with an array of five arginine residues is highly basic. The beginning of the *lydD* gene overlaps with the end of *lyz* and the 3′ moiety of *lydD* overlaps with the beginning of a small gene of unknown function, designated as *lydE*.

The spanin genes of P1, *ppfA* (o-spanin) and *mlp* (i-spanin) partially overlap and are located in yet another unlinked operon. They were identified based on the predicted features of their products, but their function has not been verified experimentally [27]. In laboratory cultures grown with shaking, it is not essential for lysis.

The organization of the known and putative cell lysis-associated genes of P1 is conserved in a P1-related phage, P7 (GenBank acc. no. AF503408). The amino acid sequences of their products, except that of LydC, are identical or nearly identical in these two phages.

The complexity of P1/P7 lytic functions poses a question about its biological sense. In this work, we dissect the functions of known and predicted P1 holins, show the different impacts of particular holins on the cell lysis of different P1 hosts, and the concerted contribution of all of them to efficient lysis in a programmed time. Additionally, we demonstrate that LydD is a pinholin, and that it has a dominant influence on the morphological transitions of *E. coli* and *Enterobacter cloacae* cells during lysis mediated by P1.

## 2. Results

### 2.1. Influence of LydD on the Phenotype of Cells Producing the P1 Lyz Endolysin

The insertional inactivation of the P1 *lyz* gene in a P1 variant encoding the thermosensitive phage repressor (C1-100) results in the inability of lysogens to lyse upon the thermal induction of P1 lytic development (Appendix A). This is consistent with the inability of the P1 amber mutant in the region of *lyz* to cause lysis of the host cell [43] and indicates that the Lyz endolysin is the only P1 muralytic enzyme required for P1-mediated cellular lysis. As a SAR domain-containing endolysin, Lyz can cause lysis in the absence of holins and can therefore be cloned only under repression control. This was achieved by cloning it under the control of the *Serratia marcescens trp* promoter with the *trp* repressor, and by growing cells in the presence of glucose.

As shown in Figure 2, P1 encodes at least three holin-like proteins, and we undertook a study to clarify their roles. The induction of the expression of *lydA* and *lyz* genes cloned in a plasmid was reported to cause *E. coli* cell lysis in the absence of any other P1 protein [42]. Thus, LydA has been described as the primary Lyz-associated holin. This seemed odd, since SAR-domain lysins require a pinholin, which is classically encoded adjacent to the lysin. There is a putative pinholin gene, *lydD*, whose 5′ end overlaps the 3′ end of the *lyz* gene [40]. Indeed, we observed that cells containing a plasmid with the entire P1/P7 *lyz* operon (*lyz*^+^*lydD*^+^*lydE*^+^) under the control of an inducible promoter lysed faster and more efficiently upon the induction of this promoter than cells containing only the *lyz* gene in an isogenic plasmid (Figure 3). The presence of *lydE* did not influence the lysis phenotype, indicating that LydD protein alone can cooperate with Lyz in cell lysis in the absence of LydA and other P1 proteins.

### 2.2. Lethal Effect of Cloned lydC or lydA on E. coli Cells

In the P1 genome, the *lydA* gene is immediately preceded by *lydC*, whose product has features characteristic of class II holins, as does LydA [40]. Our attempts to clone the *lydC* or *lydA* gene in the ColE1-derived expression vector (pGBT30) led to the formation of only a few small colonies of transformants. They either did not form colonies upon passaging or could be transferred to a liquid medium but stopped growing upon the induction of cloned gene expression. The growth cessation and lethal phenotypes of cells producing LydC or LydA observed here are consistent with the previously observed phenotypes of bacterial cells expressing cloned holin genes (see, e.g., [38,49]). They were attributed to the holin-mediated cell membrane puncturing or disruption of the proton motive force, resulting in cell death.

### 2.3. Effects of LydA, LydC, or LydD Depletion on P1-Mediated Cell Lysis

To test the contributions of LydA, LydC, and LydD to the P1-mediated lysis of *E. coli* cells, we introduced to the P1 genome, by recombinational replacements, the kanamycin resistance cassette or indels inactivating the *lydA*, *lydC*, or *lydD* gene. Additionally, P1 variants that carried the combinations of mutations in any two of these genes were constructed. The *lydA* mutant and the double *lydA lydC* mutant formed plaques on a layer of indicator strain cells (*E. coli* N99) when plates with infected cells were incubated at 42 °C (under conditions non-permissive for the formation of lysogens). In the case of the double *lydA lydD* mutant, no plaques could be observed. This shows that LydD alone is sufficient to cooperate with Lyz in cell lysis by P1, but LydC alone is not, at least under the conditions of our experiment.

In liquid medium phages deprived of functional LydC or LydD, or both these proteins could lyse *E. coli* cells, although with a delay. The delay differed slightly in the case of two different *lydD* mutants: *lydD*::kan^R^ and *lydD**Δ16-81* (Figure 4A). In contrast to that, phages deprived of LydA did not lyse cells, at least for the duration of the experiment. Taken together, our results demonstrate that LydA, LydC and LydD contribute to the P1-mediated *E. coli* cell lysis in a programmed time under these conditions. However, while LydA appears to be the main holin involved in P1-mediated cell lysis, in liquid medium, the functions of LydC and LydD are accessory.

The inability of LydD to cause lysis by P1 depleted of functional *lydA* and *lydC* genes, in the case of cells grown in liquid medium, is surprising in a view of the ability of LydD to cause lysis with the participation of Lyz in the absence of other P1 proteins (see Figure 3). It is also surprising in a view of the ability of P1 *lydA lydC* mutant to form plaques. Most likely, the contribution of LydD to cell lysis by P1 is influenced by the presence of other components of the P1 cell lysis system and by cell growth conditions.

To see whether the contributions of different holins to the cell lysis of different hosts of P1 vary, we lysogenized, with P1 or its mutants, cells of *Enterobacter cloacae* (Figure 4B). The onset of *E. cloacae* lysis upon the induction of P1 lytic development occurred later than that of *E. coli*, but at a similar optical density of the cell culture. The inactivation of *lydC* or insertional inactivation of *lydD* delayed the lysis, as in the case of *E. coli*. The deletion within the N-terminal moiety of *lydD* did not delay the onset of lysis, but slightly slowed down the dynamics of lysis. The mutation in *lydA* delayed the onset of lysis for longer than the mutations in *lydC* and *lydD*, but did not prevent lysis for the duration of the experiment, unlike in *E. coli*. Taken together, these results indicate that, although LydA, LydC, and LydD contribute to P1-mediated *E. cloacae* and *E. coli* cell lysis in a programmed time, the contribution of each one to lysis can vary depending on the bacterial growth conditions and the host of P1.

### 2.4. Influence of the P1 lyz Gene Replacement with the λ R Gene on P1-Mediated Cell Lysis

In certain phages that encode SAR-endolysins, the onset of lysis is associated with the action of pinholins, and typically the SAR-endolysin-encoding genes of these phages are in the same operons as the genes for the accompanying pinholins [11,12,13,16,19]. Thus, we tested whether the P1 *lydD* gene could encode a pinholin. P1 bacteriophage mutants with the *lyz* gene replaced with the phage λ *R* gene (encoding a canonical endolysin) and with various holin genes inactive were tested for the ability to lyse cells (Figure 5A,B). The replacement of *lyz* with *R* delayed the lysis of *E. coli* cells. The lack of functional *lydD* did not influence the lysis by the replacement mutant, suggesting that LydD protein does not contribute to lysis with the participation of R. No lysis could be seen in the case of replacement mutant lacking functional *lydA*, independent of whether *lydC* and *lydD* genes were intact or mutated. This conforms to the function of LydA as a canonical holin and is consistent with the essential role of LydA in lysis of *E. coli* cells by P1 in liquid cell cultures. The lack of functional *lydC* in the replacement mutant slightly accelerated the lysis, suggesting a possible role of LydC in the regulation of lysis timing.

Contrary to *E. coli*, the *lyz* to *R* replacement did not have a significant influence on the onset of the lysis of *E. cloacae* cells (Figure 5C,D). However, as in *E. coli*, the lack of functional *lydD* did not significantly influence the lysis by the replacement mutant, suggesting that LydD protein does not contribute to lysis with the participation of R. In support of that, the *E. cloacae* lysogen of the replacement mutant deprived of LydA, or LydA and LydC did not lyse upon the induction of P1 lytic development, even though LydA is not essential for the P1-mediated lysis of *E. cloacae* in liquid cultures, and even though the *lydD* gene was intact in the mutant. Clearly, LydD alone cannot cooperate with R endolysin in cell lysis, as if it was not a canonical holin but a pinholin.

The main protein cooperating with λ R in the lysis of *E. cloacae* cells grown in liquid medium appeared to be LydA. The replacement mutant lacking functional *lydA* did not lyse cells independent of whether the *lydC* and *lydD* genes were present or not. Moreover, the lack of functional *lydC*, or *lydC* and *lydD* in the replacement mutant only delayed the lysis. As the lack of LydD alone did not influence the lysis by the replacement mutant, this delay indicates that LydC can also contribute to lysis with the participation of R, but its role in lysis under these conditions is accessory. Surprisingly, we observed that the replacement mutant lacking *lydA* could form plaques on the layer of *E. cloacae* cells when plates with the infected cells were incubated at 42 °C, as if LydC or LydD, or two of them would be sufficient to support the function of R under these conditions.

### 2.5. Influence of LydD on E. coli Cells Producing R Endolysin

In contrast to canonical holins, which form large non-specific pores in a cytoplasmic membrane, pinholins can form only nanometer-scale pores, which are too small to allow for the passage of canonical endolysins. Since λ R is a canonical endolysin, it would not be expected to pair with LydD, if LydD is a pinholin. To test this, we used a two-plasmid system in which *R* and *lydD* genes were cloned separately in different plasmids under the control of different inducible promoters (Figure 6). The induction of the expression of each gene separately or both together caused a significant retardation in the increase in optical density of cell cultures, but no decrease in the optical density that would be characteristic of lysis. However, the number of colony-forming units in the cultures of cells with the induced expression of *lydD* decreased, indicating cell death. Clearly, LydD cannot cooperate with the canonical R endolysin in cell lysis, but can cause cell death, which supports its function as a pinholin.

### 2.6. Morphological Transitions during E. coli and E. cloacae Cell Lysis Mediated by P1 or Its Mutants

The differences in the mechanisms of canonical and pinholin–SAR-endolysin-mediated lysis were shown to be associated with differences in the morphological transitions of lysing cells [29]. Thus, we visualized *E. coli* cell lysis mediated by P1 or its *lydC* or *lydD* mutants by time-lapse microscopy, and compared it with that mediated by ʎ phage. A total of 2 out of 50 individual ʎ-mediated lysis events captured in samples of 10 independent cultures are shown in Figure 7A,B as time-lapse series (see also Appendix A). Typically, the morphological changes of *E. coli* cells during ʎ-mediated lysis started from cell shortening, which was associated with the formation of a ball-shaped form at one end (45/50 events, Figure 7A) or in the center of the cell (5/50 events, Figure 7B), and by localized cell rupture. In the case of ʎ, this rupture is caused by holins, which create large holes in the cell membrane, leading to the formation of a transient non-refractile ghost that retains a circular shape, and to the expulsion of the cytoplasmic content, as was observed previously [29]. The morphological changes of *E. coli* cells during P1-mediated lysis appeared to be different from those observed in the case of λ; they progressed through the reduction in cell length and increase in cell diameter (68/70 events) (Figure 7C,D; see also Appendix A), as was observed previously in the case of pinholin–SAR-endolysin-mediated lysis [29]. Surprisingly, the inactivation of *lydC* and *lydD* genes (Figure 6E,F) resulted in a switch in morphological changes during lysis to those characteristic of ʎ-mediated lysis: cell was shortened at one end (56/69 events, Figure 7E) or in the center (13/69 events, Figure 7F; see also Appendix A). Taken together, these results demonstrate that LydA is a canonical holin, and support the pinholin function of LydD. Additionally, they indicate that pinholin–SAR-endolysin mediated morphological transitions are dominant during the lysis of *E. coli* cells at the end of P1 lytic development.

The P1-mediated lysis of *E. cloacae* cells progressed through similar morphological changes to those that accompanied *E. coli* cell lysis: the cells shortened and thickened, converting into a more ball-shaped form, and then cell envelopes disintegrated more or less evenly (52/52 events, Figure 8A; see also Appendix A). In contrast, *E. cloacae* cell lysis caused by phage P1 with inactivated *lydD* and *lydC* genes was accompanied by slight cell shortening, the formation of a small protrusion at one cell pole, and the bursting of the end of this protrusion, resulting in the abrupt expulsion of cell contents through it (24/25 events, Figure 8B,C; see also Appendix A). In this case, as in the case of *E. coli* with the relevant mutant, the lysis occurred from one cell pole. Despite that the changes in *E. cloacae* and *E. coli* morphology at this pole during lysis were significantly different, in both cases the lysis was initiated from one cell pole, fitting to the canonical holin–endolysin-mediated pattern of cell lysis. This is consistent with the canonical holin function of LydA. Additionally, as in the case of *E. coli*, the comparison of morphological transition patterns during the lysis of *E. cloacae* cells with wild-type P1 and P1 *lydC lydD* mutant indicates that lysis mediated by wild-type P1 involves the concerted action of all P1-encoded holins/pinholins, and that pinholin-SAR-endolysin-mediated morphological transitions are dominant during *E. cloacae* cell lysis.

### 2.7. Influence of P1 Lyz Gene Replacement with λ R on Cell Morphological Transitions during Lysis

To see whether the replacement of the *lyz* gene in the P1 genome with λ *R* gene would switch the pattern of morphological transition of the replacement mutant carrying cells during lysis, we visualized lysing *E. coli* and *E. cloacae* cells with the replacement mutant. As expected, in both cases, the pattern of morphological changes during lysis switched to that characteristic of canonical holin–endolysin-mediated lysis (Figure 9). A similar switch was observed independent of whether the mutant could produce LydC, consistent with our previous observations that LydA and LydC act as canonical holins and that the pattern of morphological changes during the lysis of *E. cloacae* cells by P1 is influenced by the concerted action of LydA, LydC, and LydD.

## 3. Discussion

The lysis of a phage-infected cell, as soon as the mature phage progeny is ready to be released, is crucial for the phage’s evolutionary fate in the world of phages competing for the same hosts. The breakage of the cytoplasmic membrane and cell wall integrity is essential for lysis under any bacterial growth conditions. Most phages, such as, e.g., λ, can relay in these processes on a single pair of proteins, one of which (endolysin or SAR-endolysin) can digest cell wall peptidoglycan, while the other (holin or pinholin) can kill the cell by disrupting the transmembrane proton motive force and enabling or facilitating the access of endolysin/SAR-endolysin to the cell wall in an active form (reviewed in [1,9,22]). The analysis of the P1 bacteriophage genome indicates that its lytic system is more complex. In addition to SAR-endolysin (Lyz), holin (LydA), and antiholin (LydB), whose functions in lysis were confirmed previously [42], P1 encodes two additional proteins of amino acid sequence features characteristic of holins (LydC and LydD) [40]. In this paper, we show that LydC and LydD contribute to P1-mediated cell lysis, and that, while LydA and LydC function as canonical holins, LydD functions as a pinholin. Moreover, both holins and the pinholin are essential for the efficient lysis of bacteria in a programmed time. We demonstrate the biological sense of such lytic system complexity by showing the different contributions of particular holins/pinholins to the lysis of cells grown under different conditions and cells of different P1 hosts, *E. coli* and *E. cloacae*. Additionally, we demonstrate for the first time that, while the morphological transitions of the cells of these various hosts lysed by P1 are similar, they may differ if lysis is supported by an incomplete set of holins/pinholins.

The redundancy of the genes that could encode proteins of predicted holin features is not unique for P1 and its relatives [49,50,51,52,53]. However, whether one of these predicted holins in these cases is not an antiholin (a holin antagonist) was verified only in a few cases. For instance, the endolysin gene of *Actinomyces naeslundii* phage Av-1 is flanked by two genes that encode two functional holins of unrelated amino acid sequences [54]. In the case of *Bacillus* phage SPP1, which also encodes two holin-like proteins, cell lysis could only be caused by the concerted action of the endolysin of this phage with both of these proteins [49,53]. The results of our studies on P1 presented in this paper demonstrate for the first time the concerted action of three proteins of holin features, namely two holins and one pinholin, in phage-mediated cell lysis. Additionally, as P1 Lyz protein is the only endolysin encoded by P1, all these proteins cooperate with Lyz in cell lysis.

Typically, in the lytic modules of phage genomes, genes for canonical endolysins adjoin the genes for holins, while genes for SAR-endolysins adjoin the genes for pinholins [10,11,12,13] (reviewed in [38,53,55]). The amino acid sequences of holins and pinholins are much alike [22]. They are both characterized by the presence of transmembrane domain(s) and positively charged amino acid residues at the C-terminal end. However, the differences in the number and size of pores formed by holins (a few near-micrometer-scale pores) and pinholins (a multitude of nanometer-scale pores) are associated with phenotypic differences in lysis with the action of these proteins [17,18,19]. First, while holins can permeate the membrane for the unspecific passage of even large proteins, hence can cooperate with canonical endolysins in cell lysis, pinholins cannot support the lysis of cells producing canonical endolysins [19], as the pores that they form are too small for the unspecific passage of the latter to the periplasmic space. Second, transitions in cell morphology during lysis caused by a canonical endolysin and holin and by a SAR-endolysin and pinholin differ substantially [29]. In this paper, we show that, while LydA and LydC can support the lysis of cells by P1-producing canonical endolysin R instead of Lyz, which is characteristic of holins, LydD cannot. The latter result was confirmed by the lack of lysis of cells with the *R* and *lydD* genes cloned in separate plasmids under the control of different inducible promoters. Clearly, the inability of LydD to support the lysis of cells producing the canonical endolysin is not associated with the action of any P1 protein that could potentially inhibit the function or synthesis of LydD. Taken together, these results indicate that LydD is a pinholin, consistent with the location of the *lydD* gene in the same operon as *lyz*.

The morphological transitions of *E. coli* cells containing wild-type P1 or its mutants during lysis conform to the functional assignment of LydD to pinholins, and LydA and LydC to canonical holins, and the concerted action of all of them in P1-mediated cell lysis. While the lysis of *E. coli* by wild-type P1 progressed through morphological changes characteristic of SAR-endolysin–pinholin-mediated lysis, lysis by the P1 mutant deprived of LydD progressed through changes characteristic of canonical endolysin–holin-mediated lysis.

Comparisons of the sequence and predicted structure of LydD with the sequence and structure of prototypical pinholin S^21^68 of lambdoid phage 21 indicate similarities as well as differences. Both, LydD and S^21^68 are in about 80% alpha-helical. They both contain two TMDs and positively charged C-terminus [40,56,57]. The best matching proteins with a structure similar to the predicted structure of S^21^68 and LydD and identified with HHPred [58,59] in the PDB database are involved in ion transport. MerF, an 81-aa bacterial mercury transporter, is the best matching structural homolog of S^21^68 (2MOZ_A; 90.6% probability, matching region of S^21^68: pos. 2-65), while the 91-aa MrpF subunit of multisubunit Na+/H+ antiporter Mrp is the best matching structural homolog of LydD (6HWH_X; 59.85% probability, matching region: pos. 15-72). MerF is also among the predicted structural homologs of LydD, but it is only in the fourth position among other homologs (37.63% probability, matching region: pos. 12-76). However, while the 5-aa N-terminal region preceding the TMD1 of S^21^68 contains only one positively charged amino acid residue and the addition of two positively charged amino acid residues to it converts modified S^21^68 to a pinholin antagonist that inhibits lysis triggering [18], the 12-aa N-terminal region of LydD preceding TMD1 contains four positively charged amino acid residues [40]. Additionally, the glycine zipper motif identified in the TMD2 of S^21^68 as involved in interhelical interaction in the pathway of pinhole formation [60] is absent from LydD (data not shown). Thus, the exact mechanisms of the pinhole formation pathway by LydD and S^21^68 may differ as do the mechanisms of activation of LydD and S^21^68 cognate SAR-endolysins (reviewed in [22,56]).

Lyz was shown previously to cooperate with LydA in cell lysis when expressed with LydA from a plasmid [42]. However, when the Lyz protein is overproduced in *E. coli*, it causes cell lysis in the absence of holins, if it can be transferred to the periplasmic side of the CM with the help of SecA [5]. Thus, the ability of P1 deprived of *lydA* to retain some activity in *E. coli* cell lysis was explained previously as attributed to a slow, spontaneous release of the membrane-tethered Lyz. Our results indicate that it can be attributed, at least in part, to the activity of LydD. LydD alone could cooperate with Lyz in the lysis of *E. coli* cells with cloned *lyz* and *lydD* genes, and in the lysis of immobilized *E. coli* cells by P1. Apparently, under the latter conditions, the canonical Lyz/LydD pairing is more efficient in P1-mediated cell lysis than Lyz/LydA pairing.

In a liquid medium, LydD appears to have only an accessory role in P1-mediated lysis. LydA alone could pair with Lyz in lysis under these conditions, suggesting the sufficient production of Lyz in its active form. In its ability to cooperate with pinholin as well as with canonical holin, Lyz resembles the SAR endolysin of lambdoid phage 21, which normally cooperates with pinholin in cell lysis, but could also complement the lysis defect of an induced lambda lysogen encoding a canonical holin and depleted of its canonical endolysin gene [19].

Our results indicate that whether the Lyz–LydD or Lyz–LydA pairing predominate during lysis mediated by P1 depends on cell growth conditions and a host of P1. The P1-mediated lysis of *E. coli* cells in a liquid culture with shaking depended mainly on LydA, but LydA was not as important for the lysis of *E. cloacae* cells under similar conditions. While P1 producing LydD or LydC could not cause the efficient lysis of *E. coli* cells if it did not produce LydA, it could cause the delayed but efficient lysis of *E. cloacae* cells. Additionally, while LydA appeared to be a dominant cytoplasmic membrane puncturing protein, which could not be replaced by LydD, when bacteria grew with shaking in liquid culture, LydD was sufficient for the lysis of cells immobilized in a solid medium.

Perhaps, whether LydA or LydD predominantly cooperate with Lyz under the given conditions of P1 development depends on the regulation of their activity or the regulation of their gene expression. One possible regulator is the LydB protein, which acts as an antiholin, antagonistic to LydA. Phages carrying an amber mutation in *lydB* lysed cells prematurely [43]. Cells with a plasmid carrying the *lyz*, *lydA* and *lydB* genes cloned under the control of an inducible promoter started to lyse much later upon the induction of this promoter than cells containing the isogenic plasmid without *lydB* [42]. How exactly LydB acts is unknown, as it has no homologs among proteins of known function, and, unlike the antiholin of lambda phage, it does not contain any predicted transmembrane domains. Whether LydB can interact with LydC or LydD has not been studied.

The *lyz* and *lydD* genes, and the *lydA* gene with its cognate antiholin gene *lydB*, are expressed from late P1 promoters, opposite to *lydC*, which is preceded by a predicted early promoter [40,42]. The role of LydC in *E. coli* cell lysis by P1 appears to be an accessory one independent of the bacterial growth conditions that were tested in this study. The P1 *lydA lydD* mutant could not form plaques. Additionally, the lack of LydC did not influence cell lysis in the absence of LydA. One cannot exclude that the level of early *lydC* expression may predetermine the ratio of LydA to LydD in P1-mediated lysis. A candidate for an additional regulator might be LydE, whose predicted gene partially overlaps the end of *lydD*, which could suggest antipinholin function. However, the removal of *lydE* from the *lyz lydD lydE* operon cloned under the inducible promoter in a plasmid did not influence lysis upon the induction of transcription from this promoter (Figure 3).

Differences in the onset of lysis upon the induction of wild-type P1 and *lydA*, *lydC* or *lydD* mutants indicate that each of the P1 holin/pinholin influences lysis timing, which is a parameter of crucial importance for phage fitness. The optimal lysis time that ensures the highest phage fecundity varies even in the case of the same host strain infection by a phage, depending on the ecological settings [22,61,62]. Bacterial growth conditions and host cell density are only some examples. For instance, while delaying the onset of lysis, which results in an increased burst size, benefits a phage in environments with a low host concentration, accelerating the onset of lysis can be advantageous in environments with a high host concentration by increasing the phage population growth through a rapid establishment of new infections by progeny phages. Unlike λ, P1 is a bacteriophage with a wide host range among *Enterobacteriaceae* [39,41]. The differences between its different hosts that determine adsorption and infection efficiency, burst size and the duration of latent period may demand the more flexible regulation of lysis timing than in the case of narrow-host-range phages. Additionally, while membrane composition and permeability may not be constant even in the bacteria of the same species exposed to different environmental conditions, in the case of bacteria of different genera it may differ substantially, as implied, e.g., from differences between *Enterobacteriaceae* strains of various genera in their susceptibility to different antibiotics, antimicrobial peptides, and detergents as well as in the amounts of various compounds released upon osmotic shock (see, e.g., [63,64,65,66,67,68]). Thus, it is not surprising that the ability to permeabilize or puncture the CM of representatives of different bacterial genera efficiently and in a programmed time may require more functions and more flexible regulatory mechanisms than the ability to permeabilize or puncture the CM of more closely related bacteria. Conceivably, the contribution of two holins and one pinholin to cell lysis by P1 helps in the choice of optimal lysis timing as well as in the efficient lysis of cells grown under different conditions and originating from different hosts.

Pinholin–SAR-endolysin-encoding gene pairs were suggested to represent an intermediate stage in the evolution from the most ancient dedicated lysis systems represented by SAR-endolysins, to the holin–endolysin systems, which are considered the most advanced [19]. SAR-endolysins alone ensure slow, gradual lysis. Their cooperation with pinholins enables the lysis to be saltatory and regulated in time, but their spontaneous activation makes this regulation imprecise. In the classical endolysin–holin-mediated lysis, cytoplasmic endolysin can be accumulated in a cell in great excess without consequences, making the lysis triggering dependent only on holin activity. It is unclear whether the lytic system of P1 represents simply an evolutionary intermediate between the SAR-endolysin–pinholin system and the endolysin–holin system. In this scenario, the *lydA* and *lydC* genes could have been acquired by a P1 ancestor that initially encoded only Lyz and LydD, but any canonical endolysin gene have not been acquired or evolved yet. However, our database search for close homologs of P1 cell lysis-associated proteins indicates that the redundancy of holin/pinholin functions, while retaining SAR-endolysin, may be evolutionarily beneficial for *Punavirus* genus phages, represented by P1. It is conserved not only in P7 phage (see Figure 1), but also in other P1-related phages isolated from various bacteria (GenBank acc. no.: AF503408.1, MZ188976, MH160767; data not shown) [69,70,71,72]. Possibly, under conditions of competing with numerous phages for different hosts, e.g., in gut environments of enteric bacteria, redundant holins/pinholins could allow lysis triggering in the optimal time while SAR endolysin could additionally ensure premature lysis triggering and release of even a few phage progeny in response to any temporal membrane depolarization caused by a superinfecting phage, which might otherwise overwhelm the original infection [22]. If the premature lysis is too early to allow the primary phage maturation, at least it can eliminate the superinfecting phage and hence prevent the production of its progeny.

## 4. Materials and Methods

### 4.1. Bacterial Strains, Bacteriophages, and Plasmids

Experiments to test the lytic phenotypes of bacteriophages or their mutant derivatives were performed with the use of *Escherichia coli* N99 (*galK2*, *str*) [73] and *Enterobacter cloacae* subsp. *cloacae* DSM6234 strains (Deutsche Sammlung von Mikroorganismen und Zellkulturen GmbH) as phage hosts. *E. coli strain* DJ125 (*supE44*, *hsdR*, *thi-1*, *trh-1*, *leuB6*, *lacY1*, *tonA2*, *recA54*), which is a recombination-deficient derivative of C600 [74,75], was used for plasmid propagation and cloning experiments. Recombinational replacement experiments to transfer mutations from plasmids to bacteriophages were performed with the use of *E. coli* N99 strain. The bacteriophage P1 used in this study is a mutant (P1 *c1*-*100* IS*1*::Tn*9 mod749*::IS*5*) that carries transposon Tn*9* with a chloramphenicol resistance marker and the *c1*-*100* mutation conferring thermosensitivity to the phage repressor protein [40]. The thermosensitivity of phage repressor allows synchronous induction of P1 *c1*-*100* Tn*9* lysogens upon transfer from 30 to 42 °C. The derivatives of P1 carrying mutations in genes encoding known or predicted lytic functions and constructed in this study are listed in Table 1. The bacteriophage λ*cI*_857_, which was used as a control and as the source of *R* gene, was kindly provided by Anna Bębenek.

The plasmids used are derivatives of pMB1, pACYC184, or pSC101, and are listed in Appendix A. Nucleotide sequences of plasmid inserts that were obtained by amplification were verified by sequencing. Primers that were used for the PCR amplification of the desired fragments or for sequencing are listed in Appendix A. Oligonucleotide synthesis and sequencing services were provided by the Laboratory of DNA Sequencing and Oligonucleotide Synthesis of IBB PAS.

### 4.2. Bacterial Growth Conditions and Bacteriophage Propagation

Bacteria were grown in Luria–Bertani liquid medium (LB; Difco) or on LB medium solidified with 1.5% agar at 30, 37, or 42 °C, as indicated. When required, the medium was supplemented with glucose or maltose to a final concentration 0.2%, or with antibiotics: ampicillin (amp) (50, 100, or 500 μg/mL), chloramphenicol (cm) (12.5 or 25 μg/mL), or kanamycin (km) (12.5 or 25 μg/mL). Incubations of liquid cultures were performed with shaking (200 rpm) overnight or until the desired optical density (OD_600_). Bacteriophages were propagated upon thermal induction of lysogens. Briefly, cells lysogenized with λ*cI_857_* or P1 *c1*-*100* IS*1*::Tn*9 mod749::IS5* (or its mutant derivatives) were grown in LB or LB supplemented with chloramphenicol (12.5 μg/mL), respectively, overnight at 30 °C. The overnight cultures were diluted 50× in fresh LB with 0.2% maltose or 0.2% glucose and chloramphenicol, respectively, and grown at 30 °C until the optical density (OD_600_) reached 0.3–0.4. Prophages were thermally induced by rapid heating of cultures to 42 °C in a shaking water bath heated to about 60 °C, then the cultures were returned to 42 °C and grown with shaking for about 45–60 min until signs of lysis were noticed. The crude lysates were harvested when the OD_600_ fell to 0.15 and centrifuged at 29,030× *g* for 40 min at 4 °C. The remaining cells and cell debris were removed from the lysates by filtration through 0.22 μm syringe filter (Filtropur, Sarstedt, Nümbrecht, Germany). The titer of phages in the lysates was assayed using the standard double-layer agar method, according to Adams (1959) [76].

### 4.3. DNA Manipulation

DNA amplification, plasmid isolation, and agarose gel electrophoresis were carried out using standard techniques [77]. *E. coli* cells were transformed with plasmids as described previously [77]. Restriction enzymes, T4 DNA ligase (Sigma-Aldrich, St. Louis, MO, USA), the Klenow fragment of *E. coli* DNA polymerase I (Thermo Fisher Scientific, Waltham, MA, USA), thermostable Ex Taq DNA polymerase (Takara, Kusatsu, Japan), and DreamTaq DNA Polymerase (Thermo Fisher Scientific, Waltham, MA, USA) were used according to the supplier’s protocols.

### 4.4. Lysogenization

A portion of fresh overnight culture of bacteria (100 µL) was supplemented with 100 µL of 20 mM CaCl_2_ and 100 µL of 20 mM MgSO_4_ mixed with 100 µL of phage lysate (10^−8^–10^−9^ PFU/mL) and incubated at room temperature for 20 min to let the phages to adsorb the bacteria. Adsorption was stopped by the addition of 1 M sodium citrate (200 µL) and vigorous mixing, followed by the addition of 1 mL LB and incubation of the mixtures for 60 min at 30 °C to express the antibiotic-resistance phenotype. The phage–host mixture was then plated on solid LB medium supplemented with chloramphenicol and, when required, additional antibiotics, and incubated overnight at 30 °C to obtain colonies of lysogens. 

### 4.5. Construction of P1 Phage Mutants

All P1 *c1-100 mod749*::IS*5* IS*1*::Tn*9* mutants were constructed by RecA-mediated recombinational replacement in vivo using plasmids carrying relevant P1 genes with the desired mutations as DNA donors (see Appendix A). Cells carrying the relevant plasmid were lysogenized with phage P1 *c1-100 mod749*::IS*5* IS*1*::Tn*9*, and the lysogens were selected on LB solid medium supplemented with chloramphenicol (12.5 μg/mL) and ampicillin (100 μg/mL), and, when required, kanamycin (25 μg/mL in the case of plasmids carrying P1 genes inactivated by the insertion of kan^R^ cassette). They were purified through single colonies and used to thermally induce prophages, as described above. The obtained phages were used to lysogenize N99 cells. Lysogens expected to contain the insertions of the kan^R^ cassette in the target prophage gene were selected on LB medium supplemented with chloramphenicol and kanamycin, and then screened for resistance to ampicillin by testing their ability to grow on LB medium supplemented with chloramphenicol, kanamycin, and ampicillin (50 μg/mL). Lysogens sensitive to ampicillin were used for further studies.

Lysogens expected to contain a mutation without an antibiotic resistance determinant in the target prophage gene were selected on LB medium supplemented with chloramphenicol (12.5 μg/mL) and ampicillin (50 μg/mL). To increase the probability of recombination between the phage and the donor DNA in the plasmid, each lysogen was used to inoculate 5 mL of LB medium supplemented with chloramphenicol (12.5 μg/mL) and ampicillin (50 μg/mL). The lysogen cultures were grown overnight with shaking at 30 °C and plated on LB solid medium supplemented with chloramphenicol (12.5 μg/mL) and ampicillin (500 μg/mL). Single colonies of lysogens able to grow on this medium were used for thermal induction of prophages. The obtained phages were used to lysogenize N99 cells, and the lysogens were selected on LB solid medium supplemented with chloramphenicol only (12.5 μg/mL). Lysogens were subsequently screened for the ability to grow on medium supplemented with chloramphenicol and ampicillin (50 μg/mL), to eliminate those that contained a donor plasmid either integrated with the prophage or in a circular form. Lysogens sensitive to ampicillin were tested for the presence of the desired mutation in the prophage by PCR amplification of the relevant DNA fragment and restriction digestion or sequencing of the obtained amplicon.

### 4.6. Assays of Lysis Kinetics

A fresh overnight culture of cells lysogenized with P1 *c1-100 mod749*::IS*5* IS*1*::Tn*9* or its mutant was used to inoculate LB medium supplemented with glucose and chloramphenicol to an optical density (OD_600_) of about 0.1. Cells were grown with shaking at 30 °C. When the OD_600_ reached 0.3–0.4, the prophage was induced by thermal induction, as described above, and further changes in the optical density of the culture were measured spectrophotometrically at indicated time intervals.

### 4.7. Time-Lapse Microscopy of Living Cells

Morphological changes accompanying cell lysis upon the thermal induction of prophages in lysogens were monitored by the time-lapse microscopy of living cells. To immobilize cells in one focal plane, samples (10 µL) withdrawn from the cultures of induced lysogens before the expected lysis time were placed on a microscopic slide on the upper surface of an agarose microslab (0.7% agarose in physiological saline) and covered with a coverslip. The microslabs were prepared as described previously [78]. Cells were immediately imaged using a Zeiss Axio M2 microscope with an oil immersion objective (100×, numerical aperture = 1.4). A time-lapse video was captured at five frames per second. The video was edited and scaled using the AxioVision 4 software package (Zeiss, Oberkochen, Germany). All images were stored as .zvi files and converted to TIFF or JPEG format as necessary.

## Figures and Tables

**Figure 1 ijms-23-04231-f001:**
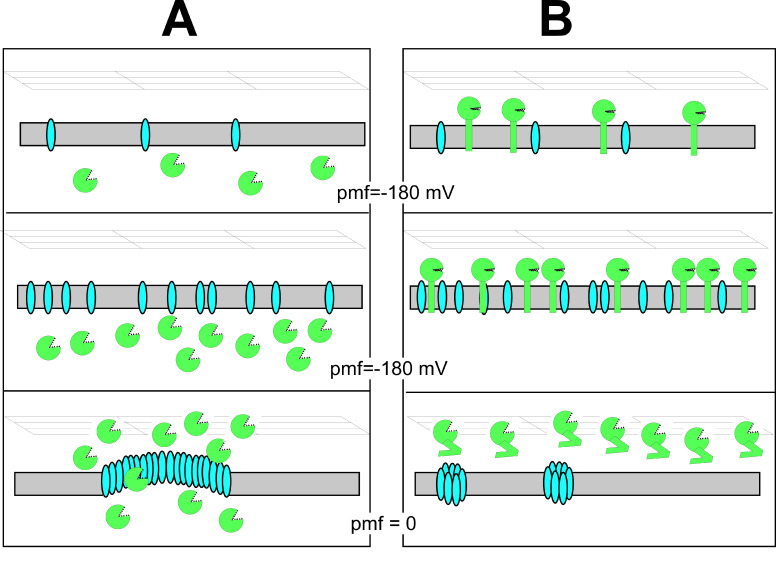
Two paradigms of murein degradation: (**A**) canonical holin–endolysin and (**B**) pinholin–SAR-endolysin pathways of murein degradation. Reproduced with permission from Young, 2014 [22] published in J. Microbiol. 2014, Springer. Only the CM (gray rectangle) and PG (grid) are shown. The cartoon series begins early in late gene expression (morphogenesis period) and progresses downwards. Holins (blue ovals in (**A**)), pinholins (blue ovals in (**B**)), soluble endolysins (green ovals with open “active site cleft”), and SAR-endolysins (green ovals with N-terminal SAR domains) depicted either in TMD conformation (green rectangle in top two panels under (**B**)) or extracted, refolded conformation (jack-knifed green rectangles in bottom panel), attached to the green globular (enzymatic and PG binding) domain. Holins accumulate in the CM (top two panels of (**A**,**B**)). Note that the prototype holin, S105 of phage λ, and pinholin, S^21^68 protein of lambdoid phage 21, accumulate as homodimers or heterodimers with their cognate antiholins (holin antagonists); however, holins are represented as single ovals here, for simplicity. The bottom panels represent the triggered cells, in which the canonical holins form a large (“micron-scale”) hole (**A**) or the pinholins form many small heptameric pinholes (**B**) in the CM. Proton motive force (pmf) of CM is indicated for each stage.

**Figure 2 ijms-23-04231-f002:**
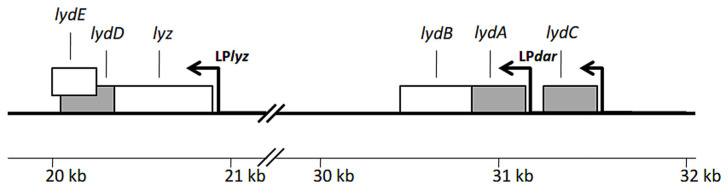
Genetic organization of the SAR-endolysin gene (*lyz*) and known or predicted holin genes in genomes of P1 and P7 phages. Known and predicted holin genes are in gray. Additional genes shown encode antiholin (*lydB*) and a putative protein of unknown function (*lydE*). Arrows indicate promoters. LP, late phage promoter. The ruler at the bottom corresponds to nucleotide positions as they appear in the published P1 genome (GenBank acc. no. NC_005856.1).

**Figure 3 ijms-23-04231-f003:**
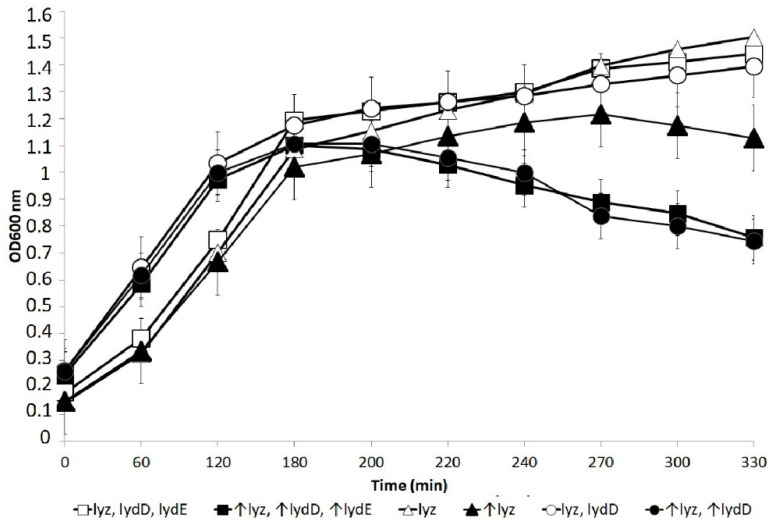
Changes in the optical density of *E. coli* cells carrying the intact P7 *lyz* operon or its truncated versions in a multicopy plasmid under the control of the *S. marcescens trp* operon promoter-operator region. Cells containing the relevant plasmid (pACE3 [*lyz*^+^l*ydD*^+^*lydE*^+^], pACE4 [*lyz*^+^*lydD*^+^] or pACE5 [*lyz*^+^]) and the pRPG18 plasmid as a source of *trp* repressor (TrpR) were grown overnight in LB medium supplemented with ampicillin and chloramphenicol at 37 °C with vigorous shaking, diluted 100× in fresh LB with the same antibiotics, and further grown under similar conditions. When the optical density of the cell cultures (OD_600_) reached about 0.2, they were supplemented with β-indole-acrylic acid (20 µg/mL) to induce transcription from the *trp* promoter. Changes in culture optical density were further monitored spectrophotometrically. Genes whose transcription from the *trp* operon promoter was induced in particular cultures are marked with arrows. The graph shows the average results of the measurement of changes in the optical density of cell cultures in at least three independent experiments. Bars represent the standard deviation.

**Figure 4 ijms-23-04231-f004:**
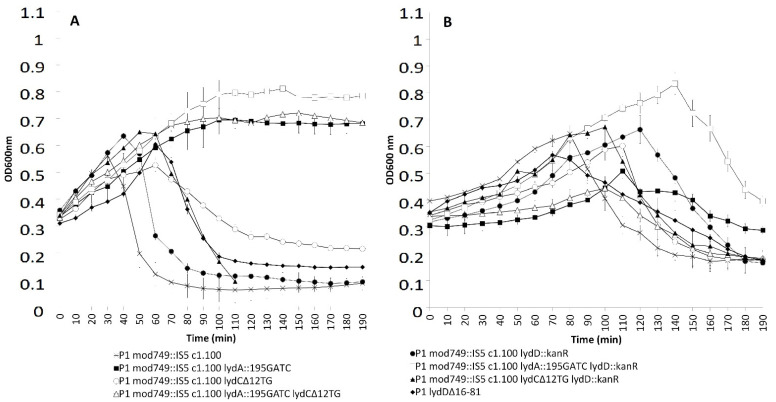
Changes in the optical density of cultures of (**A**) *E. coli* and (**B**) *E. cloacae* cells containing wild-type P1 or P1 mutants depleted of particular holin functions, upon the thermal induction of P1 lytic development. The graphs show the average results of measurement of changes in the optical density of cell cultures in at least three independent experiments. The legend applies to both charts. Bars represent the standard deviation.

**Figure 5 ijms-23-04231-f005:**
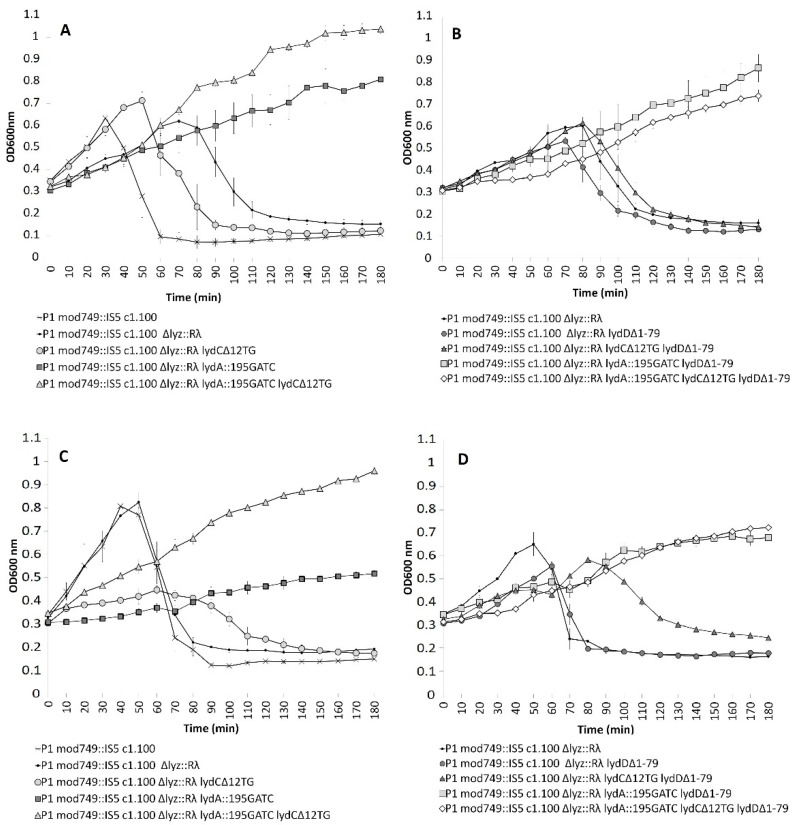
Influence of replacing the P1 *lyz* gene in the P1 genome with the λ *R* gene on the lysis of (**A**,**B**) *E. coli* and (**C**,**D**) *E. cloacae* cells, mediated by P1 or its mutants deprived of certain holin functions. The graphs show the average results of the measurement of changes in the optical density of cell cultures in at least three independent experiments. The legend applies to both charts. Bars represent the standard deviation.

**Figure 6 ijms-23-04231-f006:**
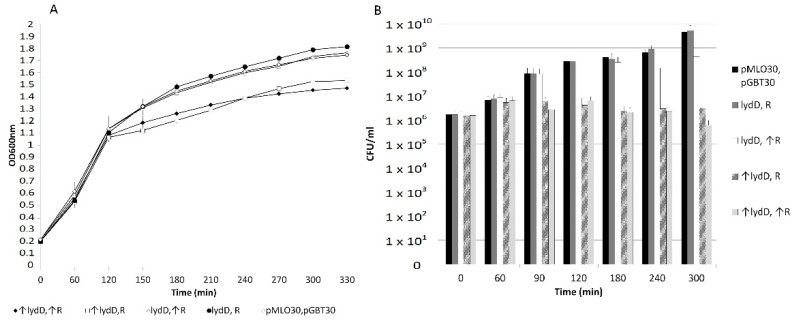
Influence of P1 LydD and λ R co-production on (**A**) *E. coli* cell growth and (**B**) ability to form colonies. Cells containing relevant plasmids (pAKI6 [*tacPO*-*lydD*^+^], pACE10 [*trpPO*-*R*^+^]) or control plasmids (pGBT30, pMLO30) were grown overnight in B medium supplemented with ampicillin and chloramphenicol at 37 °C with vigorous shaking, diluted 100× in fresh LB with these antibiotics, and grown further under similar conditions. When the optical density of cell cultures (OD_600_) reached about 0.2, they were supplemented with IPTG (1 mM) to induce the transcription of *lydD* from the *tac* promoter (time zero), grown for 60 min, and supplemented with β-indole-acrylic acid (20 µg/mL) to induce the transcription of *R* from the *trp* promoter. Changes in the culture optical density were monitored spectrophotometrically at 1 h intervals (**A**). Parallel changes in the number of colony-forming units in each culture were measured by plating diluted samples of each culture on LB agar plates supplemented with ampicillin and chloramphenicol and counting colonies after overnight incubation of plates at 37 °C (**B**). Cultures without one or both inducers added or with control plasmids served for comparative purposes. Genes, whose transcription in particular cultures was induced, are marked with arrows. Graphs show the average results of the measurement of changes in the optical density of cell cultures (**A**) or the number of colony-forming units (**B**) in at least three independent experiments. Bars represent the standard deviation.

**Figure 7 ijms-23-04231-f007:**
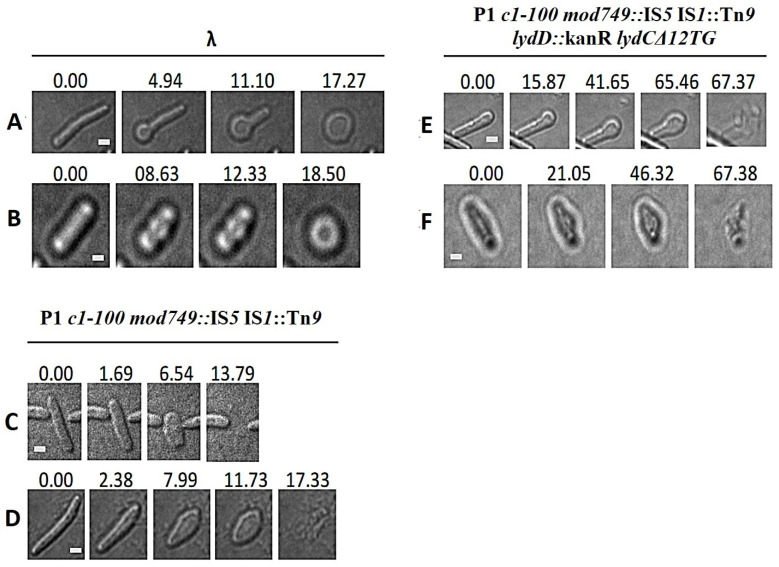
Morphological transitions of *E. coli* cells during lysis mediated by (**A**,**B**) phage λ, (**C**,**D**) wild-type phage P1, and (**E**,**F**) phage P1 mutant deprived of functional *lydC* and *lydD* genes. *E. coli* N99 lysogens were thermally induced and withdrawn from the culture just before the start of lysis to document morphological changes during lysis by time-lapse microscopy. The names of bacteriophages and their genotypes are indicated above each group of photographs. Time (in seconds) is displayed above each frame, with time zero representing the first photograph acquired before the observable morphological change or lysis. Bars = 1 µm.

**Figure 8 ijms-23-04231-f008:**
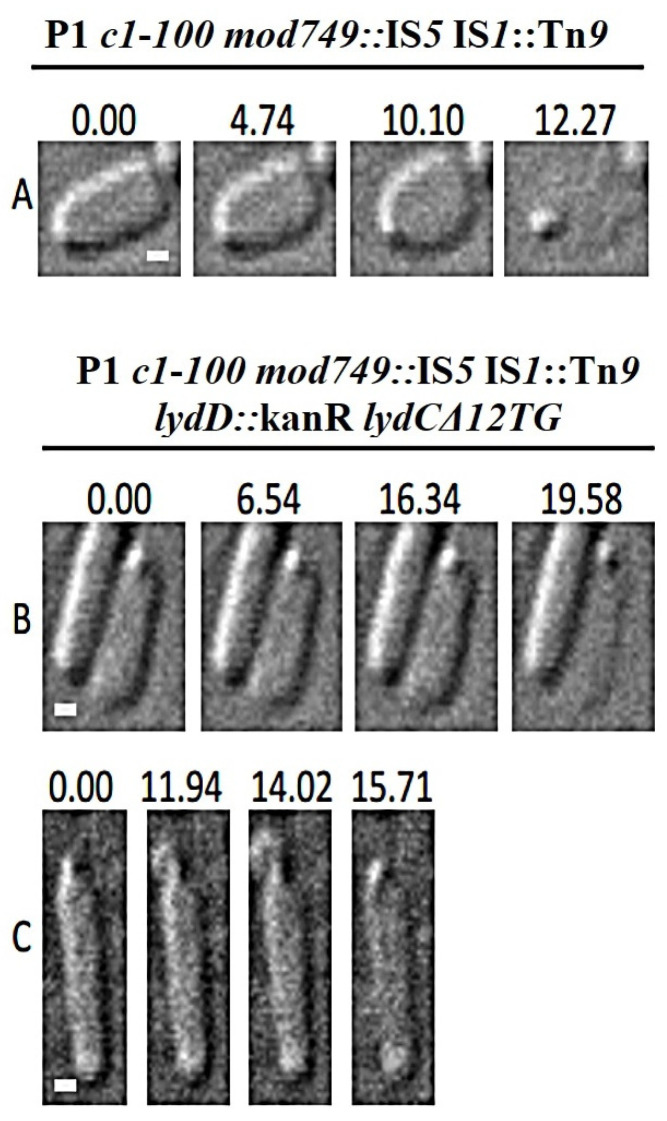
Morphological transitions of *E. cloacae* cells during lysis mediated by (**A**) wild-type phage P1 and (**B**,**C**) phage P1 mutant deprived of functional *lydC* and *lydD* genes. *E. cloacae* lysogens were thermally induced and withdrawn from the culture before the start of lysis to document morphological changes during lysis by time-lapse microscopy. The relevant genotypes of P1 phages are indicated above each group of photographs. Time (in seconds) is displayed above each photograph, with time zero representing the first photograph acquired before the observable morphological change or lysis. Bars = 1 µm.

**Figure 9 ijms-23-04231-f009:**
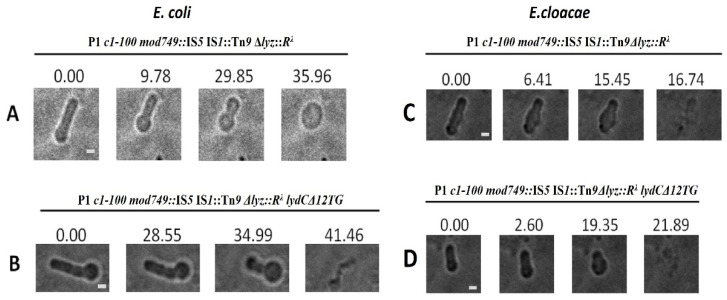
Morphological transitions of (**A**,**B**) *E. coli* and (**C**,**D**) *E. cloacae* cells during lysis mediated by phage P1 with the replacement of *lyz* gene with the λ *R* gene (**A**,**C**) and by the P1 replacement mutant deprived of functional *lydC* and *lydD* genes (**C**,**D**). *E. coli* and *E. cloacae* lysogens were thermally induced and withdrawn from the cultures before the start of lysis to document morphological changes during lysis by time-lapse microscopy. The names of bacterial species and relevant genotypes of P1 phages are indicated above each group of photographs. Time (in seconds) is displayed above each photograph, with time zero representing the first photograph acquired before the observable morphological change or lysis. Bars = 1 µm.

**Table 1 ijms-23-04231-t001:** Bacteriophage P1 *c1*-*100 mod749::*IS*5* IS*1*::Tn*9* derivatives constructed in this study.

Mutant of Bacteriophage P1 *c1*-*100* *mod749::*IS*5* IS*1*::Tn*9*	P1 Gene(s) Inactivated [Gene Inserted]	Construction; Antibiotic Resistance Marker
*lyz*::kan^R^	*lyz*	Recombinational replacement with the use of P1 *c1*-*100 mod749::*IS*5* IS*1*::Tn*9* as the recipient and plasmid pWWO2 as the DNA donor; cm^R^, kan^R^
*lydD*::kan^R^	*lydD*	Recombinational replacement with the use of P1 *c1*-*100 mod749::*IS*5* IS*1*::Tn*9* as the recipient and plasmid pAKI1 as the DNA donor; cm^R^, kan^R^
*lydCΔ12TG*	*lydC*	Recombinational replacement with the use of P1 *c1*-*100* *mod749::*IS*5* IS*1*::Tn*9* as the recipient and plasmid pAKI2 as the DNA donor; cm^R^
*lydA::195GATC*	*lydA*	Recombinational replacement with the use of P1 *c1*-*100* *mod749::*IS*5* IS*1*::Tn*9* as the recipient and plasmid pAKI3 as the DNA donor; cm^R^
*lydD::*kan^R^*lydCΔ12TG*	*lydD*, *lydC*	Recombinational replacement with the use of P1 *c1*-*100* *mod749::*IS*5* IS*1*::Tn*9 lydCΔ12_13TG* as the recipient and plasmid pAKI1 as the DNA donor; cm^R^, kan^R^
*lydD::*kan^R^*lydA::195GATC*	*lydA*, *lydD*	Recombinational replacement with the use of P1 *c1*-*100* *mod749::IS*5 IS*1*::Tn*9 lydA::195_196*GATC as the recipient and pAKI1 plasmid as the DNA donor; cm^R^, kan^R^
*lydA::195GATC lydCΔ12TG*	*lydA*, *lydC*	Recombinational replacement with the use of P1 *c1*-*100* *mod749::IS*5 IS*1*::Tn*9 lydCΔ12_13TG* as the recipient and plasmid pAKI3 as the DNA donor; cm^R^
*Δlyz::R^λ^*	*lyz* [*R*]	Recombinational replacement with the use of P1 *c1*-*100* *mod749::*IS*5* IS*1*::Tn*9* as the recipient and plasmid pAKI13 as the DNA donor; cm^R^
*Δlyz::R^λ^ lydCΔ12TG*	*lyz* [*R*], *lydC*	Recombinational replacement with the use of P1 *c1*-*100* *mod749::IS*5 IS*1*::Tn*9 lydCΔ12_13TG* as the recipient and plasmid pAKI13 as the DNA donor; cm^R^
*Δlyz::R^λ^* *lydA::195GATC lydCΔ12TG*	*lyz* [*R*], *lydA*, *lydC*	Recombinational replacement with the use of P1 *c1*-*100* *mod749::IS*5 IS*1*::Tn*9 lydA::195_196GATC lydCΔ12_13TG* as the recipient and plasmid pAKI13 as the DNA donor; cm^R^
*Δlyz::R^λ^* *lydA::195GATC*	*lyz* [*R*], *lydA*	Recombinational replacement with the use of P1 *c1*-*100* *mod749::IS*5 IS*1*::Tn*9 Δlyz::R^λ^* as the recipient and plasmid pAKI3 as the DNA donor; cm^R^
*Δlyz::R^λ^ lydD*Δ1-79 *	*lyz* [*R*], *lydD*	Recombinational replacement with the use of P1 *c1*-*100* *mod749::IS*5 IS*1*::Tn*9* as the recipient and plasmid pAKI23 as the DNA donor; cm^R^
*Δlyz::R^λ^ lydA*::195GATC *lydD*Δ1-79 *	*lyz* [*R*], *lydA*, *lydD*	Recombinational replacement with the use of P1 *c1*-*100* *mod749::IS*5 IS*1*::Tn*9 lydA::195_196GATC* as the recipient and pAKI23 plasmid as the DNA donor; cm^R^, kan^R^
*Δlyz::R^λ^ lydC*Δ12TG *lydD*Δ1-79 *	*lyz* [*R*], *lydC*, *lydD*	Recombinational replacement with the use of P1 *c1*-*100* *mod749::IS*5 IS*1*::Tn*9 lydCΔ12_13TG* as the recipient and plasmid pAKI23 as the DNA donor; cm^R^
*Δlyz::R^λ^ lydA*::195GATC *lydC*Δ12TG *lydD*Δ1-79 *	*lyz* [*R*], *lydA*, *lydC*, *lydD*	Recombinational replacement with the use of P1 *c1*-*100* *mod749::IS*5 IS*1*::Tn*9 lydA::195_196GATC lydCΔ12_13TG* as the recipient and plasmid pAKI23 as the DNA donor; cm^R^
*lydDΔ16-81* *	*lydD*	Recombinational replacement with the use of P1 *c1*-*100* *mod749::IS*5 IS*1*::Tn*9* as the recipient and plasmid pAKI26 as the DNA donor; cm^R^

* *lydE* gene and the Shine–Dalgarno sequence of *lydE* are intact in the P1 *lydD* deletion mutants.

## Data Availability

Not applicable.

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
