# Peer review of "Functional Dissection of P1 Bacteriophage Holin-like Proteins Reveals the Biological Sense of P1 Lytic System Complexity"

_ijms, 2022, doi:10.3390/ijms23084231_

Round 1

Reviewer 1 Report

General view

This is a potentially interesting MS suggesting that phage P1 with its single endolysin utilizes combinations of 4 different holins to adapt its lytic cycle to environmental conditions, host strains and host genotypes. As written, however, the paper lacks the clarity that would be achieved by placing it squarely in the context of the well-established holin/lysin paradigms. It is suggested, therefore, that the introduction begin with a clear description of the two standard holin systems, of which the authors are clearly well aware, and then showing how P1 departs from these in its adaptation to different conditions, and its use of various holins. In the classical paradigm, the endolysin is free in  the cytoplasm and the holin, dispersed in the cellular membrane, is triggered to assemble into large rafts, which create a very large site of membnrane disruption, providing the fully active endolysin with access to the cell wall peptidoglycan, resulting a local disruption of the cell envelop with escape of the cell’s contents. In a more recently described second paradigm, involving periplasmically located endolysins with SAR (signal-anchor-release) domains, the inactive endolysin anchored to the cytoplasmic membrane is released  and folded to its active form when triggered by depolarization of the membrane caused by the individual holin molecules which  make ‘pinholes’ in the membrane. Authors would be well-advised to reproduce Fig. 2 from the 2014 review by Young  (J Microbiol. 2014 Mar; 52(3): 243–258).), illustrating these two schemes. As noted by the authors, P1 immediately departs from these schemes, as the primary pairing of its (SAR-containing) endolysin Lyz, is reported to be with a canonical holin, LyzA. But, as the authors demonstrate, this situation is not so simple, since there is a putative pinholin gene, lyzD, adjacent to lyz and under certain conditions, this canonical pairing is more  efficient than the Lyz/LyzA pairing. In general, the results would be much more informative  if the authors show, in each case, how they differ from or conform to the paradigmatic cases and how this benefits the phage. The authors emphasize the effects of different genotypes on the growth rates and plateau levels of (non-lysing) cells following prophage induction. However, once the phage life cycle is induced, the cell is taken over by phage reproduction and, for non-lysing cells, the observed increase in optical density is caused by residual cellular functions and should not be regarded as growth. These measurements and the plateau levels are therefore of little or no significance and should not be emphasized.

Specific questions and comments

  1. Errors in English: Line 130: “sequences of their products, but LydC are identical or nearly identical in both these…” Do the authors mean “except for that of LydC” ? Plazmid? Bursted?
  2. Fig 2 shows that cloned Lyz/LyD/LyE can cause lysis in liquid medium. This contradicts the statement in line 203 that LydA and not LydD is responsible for Lyz-induced lysis in liquid medium.
  3. Figs 6,7,8: Times are given in seconds. Authors must mean minutes.
  4. Lines 139-150: Has it never been shown that Lyz is the only P1 lysin? If not, then the following text is suggested: “Mutational inactivation of lyz eliminated P1-induced cellular lysis, indicating that Lyz is the only muralytic enzyme required for P1-induced cellular lysis. As a
    SAR domain-containing endolysin, it could cause lysis in the absence of holins and could therefore be cloned only under repression control. This was achieved by cloning it under control of the Serratia marcescens trp promoter with the trp repressor plus glucose also present.”
  5. Lines 151-7 – very confusing. Suggested text: “As shown in Fig. 1, at least 4 holin-like proteins are encoded by phage P1 and we have undertaken a study to clarify their roles.  To begin with,  it has been reported that LyzA was the primary Lyz-associated holin (6). [authors: was lyzD present in the lyz clone described in ref 6?]. This seemed odd since SAR-domain lysins require a pinholin, which is classically encoded adjacent to the lysin. Indeed, there is, in fact, a putative  pinholin gene, lyzD, 3’ to the lyz gene [authors conclude later in the paper that LyzD is a pinholin but never perform any  direct test, or compare the sequence with known pinholins] aand we show here that the cloned lyz/lyzD complex is considerably more effective for lysis than a lyz/lyzA combination (see Fig. 2). In fact, if [or since] the lyz clone in ref 6 also contained lyzD, one would have to argue that LyzA was an  antagonist in a plaque-formation assay, even though it can act alone as a canonical holin in liquid media” [why are genes lyzB and E not included in the analysis?]
  6. Lines 173-91 –far more complex than necessary. Since authors mention that cloned class II holins are inhibitory and/or lethal, and since LyzA and C  are homologous and resemble class II holins, the detailed description of their behavior is superfluous and detracts from the flow of the paper. Also, the heading should mention both A and C.
  7. Line 197: Given the results shown above (lines 151-8 and Fig. 2, it is not at all surprising that neither LydA nor LydC is required for P1-induction of lysis. What is surprising is that LydA alone can cooperate with Lyz in liquid medium to cause lysis. This suggests that sufficient Lyz is produced in its active form without the need for a pinholin, and therefore that the two paradigms are not mutually exclusive.  Authors are asked to articulate this point.
  8. Section 2.3. A test for the effects of a deletion of lydD is essential.
  9. There is clearly a conflict between the present data and earlier reports, in that LyzD, almost certainly a pinholin, rather than LyzA, is very likely the primary Lyz-associated holin; authors must definitively address this issue.
  10. Although LyzC has holin-like properties, it appears to have at most a very minor role in P1-mediated lysis, yet is given a prominent place in the paper. This emphasis seems inappropriate.
  11. Since lambda R is a canonical endolysin, it would not be expected to pair with LyzD, if LyzD is a pinholin. The section on R vs D should be framed as a test of this, and the result indicated as supporting it.

Author Response

REVIEWER 1

General view

This is a potentially interesting MS suggesting that phage P1 with its single endolysin utilizes combinations of 4 different holins to adapt its lytic cycle to environmental conditions, host strains and host genotypes. As written, however, the paper lacks the clarity that would be achieved by placing it squarely in the context of the well-established holin/lysin paradigms. It is suggested, therefore, that the introduction begin with a clear description of the two standard holin systems, of which the authors are clearly well aware, and then showing how P1 departs from these in its adaptation to different conditions, and its use of various holins. In the classical paradigm, the endolysin is free in the cytoplasm and the holin, dispersed in the cellular membrane, is triggered to assemble into large rafts, which create a very large site of membrane disruption, providing the fully active endolysin with access to the cell wall peptidoglycan, resulting a local disruption of the cell envelop with escape of the cell’s contents. In a more recently described second paradigm, involving periplasmically located endolysins with SAR (signal-anchor-release) domains, the inactive endolysin anchored to the cytoplasmic membrane is released and folded to its active form when triggered by depolarization of the membrane caused by the individual holin molecules which make ‘pinholes’ in the membrane. Authors would be well-advised to reproduce Fig. 2 from the 2014 review by Young (J Microbiol. 2014 Mar; 52(3): 243–258).), illustrating these two schemes. As noted by the authors, P1 immediately departs from these schemes, as the primary pairing of its (SAR-containing) endolysin Lyz, is reported to be with a canonical holin, LyzA. But, as the authors demonstrate, this situation is not so simple, since there is a putative pinholin gene, lyzD, adjacent to lyz and under certain conditions, this canonical pairing is more efficient than the Lyz/LyzA pairing. In general, the results would be much more informative if the authors show, in each case, how they differ from or conform to the paradigmatic cases and how this benefits the phage.

Response

We thank the reviewer for helpful suggestions how to improve the manuscript. The introduction section was shortened and corrected, as suggested, by placing a description of phage cell lysis systems at the beginning. Figure 2 from the 2014 review by Young was included in the Introduction section to illustrate two schemes of lysis. Additionally, we removed from the introduction the mention of SP-endolysins, as irrelevant to the main subject of this study, and disrupting the emphasis on canonical holins and pinholins of phages infecting Gram-negative hosts. The discussion was reorganized and supplemented to discuss differences from paradigmatic cases observed during lysis by P1. Possible benefits for P1 from these differences are considered in various parts of the discussion section : see e.g., L. 481-591.

The authors emphasize the effects of different genotypes on the growth rates and plateau levels of (non-lysing) cells following prophage induction. However, once the phage life cycle is induced, the cell is taken over by phage reproduction and, for non-lysing cells, the observed increase in optical density is caused by residual cellular functions and should not be regarded as growth. These measurements and the plateau levels are therefore of little or no significance and should not be emphasized.

Response

We agree with this comment. The incorrect interpretation and discussion of cell culture density increase upon induction of P1 lytic development was removed from the manuscript text, and the Results section was modified, accordingly (L.211-235).

Specific questions and comments

    1.  

Errors in English: Line 130: “sequences of their products, but LydC are identical or nearly identical in both these…” Do the authors mean “except for that of LydC” ? Plazmid? Bursted?

Corrected, as requested.

    1.  

Fig 2 shows that cloned Lyz/LyD/LyE can cause lysis in liquid medium. This contradicts the statement in line 203 that LydA and not LydD is responsible for Lyz-induced lysis in liquid medium.

Figure 2 (current Figure 3) shows the results of experiment with the lyz operon cloned in a plasmid under the control of an inducible promoter, but in cells that do not produce other P1-encoded proteins, while the statement in line 203 applied to experiments with cells containing P1 depleted of single or multiple holins. It is of notion that induction of lyz and lydA genes cloned in a plasmid by Schmidt et al. (1996) also caused E. coli cell lysis, which confirms our results showing the involvement of LydA in the P1-mediated lysis of cells grown in liquid medium. The contradiction between the results of experiment shown in former Figure 2 (current Figure 3) and experiments shown in former Figure 3 (current Figure 4) and concluded in line 203 may result from yet unknown regulatory connections between different P1 operons/proteins involved in cell lysis. Possible explanation of this differences is added to the results section (L. 219-224).

    1.  

Figs 6,7,8: Times are given in seconds. Authors must mean minutes.

Times are given in correct units (seconds).

    1.  

Lines 139-150: Has it never been shown that Lyz is the only P1 lysin? If not, then the following text is suggested: “Mutational inactivation of lyz eliminated P1-induced cellular lysis, indicating that Lyz is the only muralytic enzyme required for P1-induced cellular lysis. As a
SAR domain-containing endolysin, it could cause lysis in the absence of holins and could therefore be cloned only under repression control. This was achieved by cloning it under control of the
Serratia marcescens trp promoter with the trp repressor plus glucose also present.”

We thank the reviewer for this suggestion. The suggested correction was introduced to the text. See L. 158-165.

    1.  

Lines 151-7 – very confusing. Suggested text: “As shown in Fig. 1, at least 4 holin-like proteins are encoded by phage P1 and we have undertaken a study to clarify their roles. To begin with, it has been reported that LyzA was the primary Lyz-associated holin (6). [authors: was lyzD present in the lyz clone described in ref 6?]. This seemed odd since SAR-domain lysins require a pinholin, which is classically encoded adjacent to the lysin. Indeed, there is, in fact, a putative pinholin gene, lyzD, 3’ to the lyz gene [authors conclude later in the paper that LyzD is a pinholin but never perform any direct test, or compare the sequence with known pinholins] aand we show here that the cloned lyz/lyzD complex is considerably more effective for lysis than a lyz/lyzA combination (see Fig. 2). In fact, if [or since] the lyz clone in ref 6 also contained lyzD, one would have to argue that LyzA was an antagonist in a plaque-formation assay, even though it can act alone as a canonical holin in liquid media” [why are genes lyzB and E not included in the analysis?]

We corrected the entire paragraph, as suggested (see L. 158-176). We also checked the strategy of constructing the plasmid with cloned lyz, lydA and lydB genes, that was described by Schmidt at al., 1996, and verified that this plasmid did not contain the lydD gene. The lydE gene was included in the analysis of the contribution of LydD to the lysis of cells containing the cloned lyz gene (see Figure 3 [=Figure 2 in the previous version of the manuscript]. However, it did not influence the lysis phenotype and thus it was not further analyzed. It is of notion that while the insertion of kanamycin resistance cassette in lydD in one of our mutants may have a polar effect on expression of lydE, any of the two deletions in the N-terminal moiety of lydD gene that we constructed did not distract the lydE gene and its SD sequence. As to LydB, it was shown by Schmidt et al. 1996, to function as antiholin, antagonistic to holin LydA, which is encoded by the same operon as that encoding LydB. This is mentioned in the introduction section. As our study has been focused on P1 holin-like proteins we did include LydB in experimental studies that are described in this manuscript. However, we mentioned the importance of regulatory mechanisms controlling the expression of P1 lyz, lydA, lydC, and lydD genes and the activity of their products, for the selection by P1 of optimal lysis time of any host and under any conditions (see L. 528-574).

As to LydD, the results of our analysis indicate that LydD is a pinholin: it cooperates with SAR-endolysin Lyz in cell lysis but cannot cooperate with canonical endolysin R; morphological transitions during cell lysis by P1 are similar to those that occur during SAR-endolysin-pinholin-mediated lysis. This pattern of morphological transitions depends on LydD and is switched to canonical endolysin-holin pattern in the absence of LydD. Additionally, the pinholin function of LydD conforms to the location of lydD gene in the same operon as the lyz gene. We considered in the discussion section similarities and differences between LydD and prototypical pinholin S2168 (see L. 464-483).

    1.  

Lines 173-91 –far more complex than necessary. Since authors mention that cloned class II holins are inhibitory and/or lethal, and since LyzA and C are homologous and resemble class II holins, the detailed description of their behavior is superfluous and detracts from the flow of the paper. Also, the heading should mention both A and C.

The heading of subchapter 2.2 was corrected. Additionally we abbreviated the text of this subchapter as suggested (see L. 191-199).

    1.  

Line 197: Given the results shown above (lines 151-8 and Fig. 2, it is not at all surprising that neither LydA nor LydC is required for P1-induction of lysis. What is surprising is that LydA alone can cooperate with Lyz in liquid medium to cause lysis. This suggests that sufficient Lyz is produced in its active form without the need for a pinholin, and therefore that the two paradigms are not mutually exclusive. Authors are asked to articulate this point.

We articulated this point in the discussion section, as suggested (see L. 493-498). Additionally "surprising" was removed from the relevant sentence (see L. 203).

    1.  

Section 2.3. A test for the effects of a deletion of lydD is essential.

The 5' end of lydD (3 initial codons) overlaps with the 3' end of lyz gene, and the 3' moiety of lydD (36 terminal codons) overlaps with the 5' moiety of lydE. Thus, to retain the lyz and lydE genes of the lyz operon intact, and to retain the translation start site of lydE, we deleted 22 codons internal to the N-terminal moiety of lydD (this region encodes the first transmembrane domain of lydD and a few preceding amino acid residues). The curves showing the kinetics of E. coli and E. cloacae cell lysis by the mutant P1 are included in Figure 3A and 3B, respectively, and the interpretation of results is in the text of subchapter 2.3.

We did not have the problem with deleting the entire N-terminal moiety of lydD in experiments, in which the lyz gene of P1 was replaced with the lambda R gene. Thus, in the P1 mutant used in these experiments nucleotide residues from pos. 1 to 79 of lydD were deleted. The results with N-terminal deletions in lydD basically complement the results with the insertional inactivation of lydD, although they are not identical.

    1.  

There is clearly a conflict between the present data and earlier reports, in that LyzD, almost certainly a pinholin, rather than LyzA, is very likely the primary Lyz-associated holin; authors must definitively address this issue.

We addressed this issue in the discussion section (L. 484-507). However, we also emphasize that whether holin LydA or pinholin LydD dominate in contributing to lysis depends on cell growth conditions and on the host of P1.

    1.  

Although LyzC has holin-like properties, it appears to have at most a very minor role in P1-mediated lysis, yet is given a prominent place in the paper. This emphasis seems inappropriate.

We abbreviated the discussion section concerning lydC, as suggested (see L. 517-523).

    1.  

Since lambda R is a canonical endolysin, it would not be expected to pair with LyzD, if LyzD is a pinholin. The section on R vs D should be framed as a test of this, and the result indicated as supporting it.

The text of sub-chapter 2.5 was changed, as suggested (see L. 300-310).

Reviewer 2 Report

Although the study is valuable, it has some shortcomings. Various situations should be considered that will increase the research value. The introduction and discussion should be modified with clear understanding for readers. The results should be rearranged to be more understandable.

Typos should be corrected. The article should be accepted after revision.

Author Response

REVIEWER 2

We thank the reviewer for helpful suggestions how to improve the manuscript. The introduction section was shortened and reorganized, as requested, to focus it exclusively on phage lytic functions. We also reorganized the discussion section and modified certain parts of the results section to make the manuscript easier to read and to make its content more clear.